# Quality Specific Associations of Carbohydrate Consumption and Frailty Index

**DOI:** 10.3390/nu14235072

**Published:** 2022-11-29

**Authors:** Toshiko Tanaka, Maria Kafyra, Yichen Jin, Chee W. Chia, George V. Dedoussis, Sameera A. Talegawkar, Luigi Ferrucci

**Affiliations:** 1Longitudinal Studies Section, Translational Gerontology Branch, National Institute on Aging, Baltimore, MD 21224, USA; 2Department of Nutrition and Dietetics, School of Health Science and Education, Harokopio University, 17671 Athens, Greece; 3Department of Exercise and Nutrition Sciences, Milken Institute School of Public Health, The George Washington University, Washington, DC 20052, USA; 4Clinical Research Core, National Institute on Aging, National Institutes of Health Intramural Research Program, Baltimore, MD 21224, USA

**Keywords:** carbohydrate consumption, frailty index, aging, epidemiology

## Abstract

**Background**: The quality of carbohydrate consumed may influence the risk of frailty. In this study, we tested the hypothesis that indices of carbohydrate intake are associated with trajectories of frailty in participants of the Baltimore Longitudinal Study of Aging (BLSA). **Methods**: Cross sectional and longitudinal analyses were conducted in 1024 BLSA participants to examine the association between usual intake of carbohydrate and frailty index. Seven measures of carbohydrate consumption were estimated using data derived from Food Frequency Questionnaires (FFQs) and examined in association with a 43-item Frailty Index (FI). **Results:** In cross-sectional analyses, there was a significant, positive association between higher tertiles of total carbohydrate, glycemic load, and non-whole grains and FI. Conversely, higher tertiles of fiber-to-carbohydrate ratio was associated with lower FI. These differences persisted over the follow-up period of up to 13.8 years. Women in the highest tertile of the fiber-to-carbohydrate ratio showed a less steep increase in FI over time. **Conclusions**: Carbohydrate intake was positively associated with increased frailty risk in the BLSA participants, whereas a higher fiber-to-carbohydrate ratio was related to reduced risk for frailty.

## 1. Introduction

Frailty is a geriatric syndrome associated with poor quality of life and increased risk for adverse health outcomes due to age-related decline in multiple domains [1,2]. Several operationalizations of frailty status exist, including frailty phenotype (i.e., presence of at least 3 of 5 criteria including weakness, unintentional weight loss, exhaustion, slower walking time, and low physical activity) [3], and the Frailty Index (FI) deficit accumulation model, which quantifies the presence of various age-related deficits that include diseases and health-related variables [4]. In 2008, Searle et al. outlined a method to develop a FI as a collective score including variables that increase with age but do not saturate (no early or ceiling effect) too early, including dimensions that encompasses various domains, and uses variables that are being measured consistently over time to allow the assessment of FI longitudinally [5]. 

Diet quality is an important determinant of frailty status [6]. Prior research has highlighted the importance of adequate protein intake in older populations to minimize the progressive muscle loss that comes with aging leading to sarcopenia and thereby increasing the risk for frailty [7]. In addition to protein, other macronutrients including carbohydrates, in particular added sugars, has been associated with various adverse health effects including frailty, and Alzheimer’s disease [8,9,10]. These observations have important public health relevance as recent reports indicate that the dietary habits of the American population are progressively worsening, in particular the intake of simple carbohydrates and fat increased significantly over the period 2001 to 2018 [11]. Thus, is it important to understand and characterize whether quality of carbohydrate consumption affects health trajectories in older adults. With respect to frailty, one study demonstrated associations between quality of dietary carbohydrate with incidence of frailty phenotype [12]. However, no previous study explored the relationship between dietary carbohydrate quality and frailty using the deficit accumulation definition. While frailty phenotype and frailty index are complementary measurements of frailty, they provide different information about the health status of an individual [13]. Frailty phenotype categorizes individuals into categories of frailty states, thus may capture individuals at the most severe state frailty. Frailty index is a continuous variable that places individuals on a spectrum of frailty. Thus, frailty index allows the examination of the relationship between carbohydrate consumption with frailty from the early to later phase of the frailty spectrum. 

The present study seeks to investigate cross-sectional and longitudinal associations between various indices of carbohydrate intake (including percentage of carbohydrates, glycemic load, total grains, whole grains, non-whole grains, and fiber-to-carbohydrate ratio) and frailty index in participants of the Baltimore Longitudinal Study of Aging (BLSA).

## 2. Materials and Methods

### 2.1. Study Design and Participants

The BLSA is an ongoing open cohort study of normative aging that was initiated in 1958. Detailed study protocols of the BLSA cohort have been provided elsewhere [14,15]. Briefly, the BLSA cohort includes community-dwelling men and women who reside primarily in the Washington, DC-Baltimore, MD area and who are seen every 1–4 years from study enrollment until death. At each visit, the participants undergo comprehensive assessments of physical and neurocognitive function, provide information on medical history, complete study-related questionnaires including dietary assessment, undergo laboratory and radiologic tests and other measurements during on average 3-day stay on the Clinical Research Unit of the Intramural Research Program of the National Institute on Aging (IRP, NIA) or at a home visit for the most debilitated. These analyses included 1024 participants with diet and frailty index data, and a subset of 806 with follow up data on frailty index. The study protocol (Protocol number 03-AG-0325) was approved by the National Institutes of Health Intramural Research Program Institutional Review Board and informed consent was obtained from participants at each visit.

### 2.2. Dietary Assessment and Carbohydrate Quality Indicators

Carbohydrate consumption was derived from a validated Food Frequency Questionnaire (FFQ) initiated in the BLSA in 2015. The FFQ was administered by clinic staff using paper forms and, beginning in 2016, assessment was conducted using computer-based REDCap surveys. The University of Minnesota Nutrient Data System for Research program was used to generate energy and nutrient estimates. To exclude evident outliers and possible mistakes in data entry, we included only individuals with daily energy intake between 600 and 4800 kcal [16]. We evaluated several parameters that capture the quality of carbohydrate consumption, including consumption of total energy from carbohydrates, total glycemic load, total grain (serving/day), non-whole grains (serving/day), whole grains (serving/day), and ratio of fiber (g) to carbohydrates. Glycemic load was calculated based on the following formula:Glycemic Load = Glycemic Index×available carbohydrate / 100.

### 2.3. Construction of Frailty Index (FI)

A 43-item FI was constructed following the procedures outlined by Searle et al. [5]. Items selected included, reported difficulty with 15 basic and instrumental activities of daily living (ADL/IADL: walking up 10 steps, lifting and carrying 10 lbs, getting in and out of bed/chairs, bathing and showering, dressing, eating, using the toilet, walking across a small room, doing heavy housework, preparing your own meals, shopping for personal items, using the telephone, taking medication, managing finances, urinary or fecal incontinence), self-rated health assessed using short form health survey SF-12 [17], five items from the Center for Epidemiologic Studies Depression (CES-D) scale [18] (feel depressed, feel everything is an effort, could not get going, feel lonely, and feel happy), four items of the Mini-Mental State Examination [19] (MMSE: orientation to time, orientation to place, attention, and recall), and presence vs. absence of 14 common age-related conditions (cancer, anemia, hypertension, heart disease, congestive heart failure, stroke, peripheral artery disease, COPD, chronic kidney disease, hip replacement, joint pain, depression, Parkinson’s, cognitive impairment), 5% weight loss in the past year, low physical activity (lowest quartile of physical activity in the past year), slowness (lowest quintile walking speed stratified by sex and height), and weakness (lowest quintile grip strength stratified by sex and BMI). FI was calculated for participants with less than 20% missing data at each study visit.

### 2.4. Assessment of Covariates

Participant demographic characteristics, including age and sex, were collected during a medical examination. Body mass index (BMI) was calculated as the ratio of weight (kg) and height squared (m^2^). Presence of diabetes was determined as fasting glucose ≥ 126 mg/dL or taking medication. 

### 2.5. Statistical Analysis

Differences in baseline continuous and categorical variables between men and women were assessed using one-way ANOVA and chi-square tests, respectively. Each carbohydrate consumption variable was categorized using sex-specific tertiles of intake, and the lowest tertile of carbohydrate consumption was considered as the reference group. The consumption of carbohydrate differed by sex, thus, we conducted analysis in combined as well as sex-stratified sample. For each carbohydrate quality measure, cross-sectional associations between carbohydrate quality and FI at baseline were conducted using linear regression. Longitudinal differences in trajectory of FI by baseline carbohydrate quality were assessed using the linear mixed effects model with random slope and intercept. Differences in trajectory slope evaluated the significance of the interactions between consumption and follow up time (years). When the interaction was not significant, the interaction term was removed from the model, and the main effect of carbohydrate consumption was evaluated. All regression models were adjusted for age, sex, total energy intake, BMI, diabetes, percent energy from polyunsaturated fatty acids (PUFA), saturated fatty acid (SFA), monounsaturated fatty acid (MUFA), and total fiber (except for the analysis of the fiber-to-carbohydrate ratio).

## 3. Results

### 3.1. Cross-Sectional Associations of Carbohydrate Intake with Frailty Index

The characteristics and dietary intake of the BLSA participants are displayed in Table 1. Women were significantly younger, with a lower prevalence of diabetes, had lower average fasting glucose, lower BMI, and lower intake estimates of total energy, glycemic load, total grains, whole grains, non-whole grains. On the contrary, women reported higher percentage of energy intake from carbohydrates and polyunsaturated fatty acids (PUFA). Since the carbohydrate intake pattern was significantly different by sex, both combined and sex-stratified analyses were conducted.

In the analysis of men and women combined, being in the higher tertiles of % total carbohydrates (low vs. medium, medium vs. high, low vs. high), glycemic load (low vs. high, medium vs. high), total grains (low vs. high), and non-whole grains (low vs. medium, low vs. high) was associated with higher FI (Figure 1). Conversely, being in the higher tertiles of fiber: carbohydrate (low vs. medium, low vs. high) was associated with lower FI. Similar trends were observed in the sex stratified analysis. In men, being in the highest tertiles of % total carbohydrate (low vs. high, medium vs. high), and non-whole grains (low vs. medium) was associated with higher FI while being in the higher tertile of fiber: carbohydrate consumption (low vs. medium, low vs. high) was associated with lower FI. In women, being in the higher tertile of % carbohydrate (low vs. high), glycemic load (low vs. high), total grain (low vs. high), and non-whole grains (low vs. medium, low vs. high) was associated with higher FI while being in the higher tertile of fiber: carbohydrate consumption (low vs. medium) was associated with lower FI.

### 3.2. Longitudinal Association of Carbohydrate Intake with Frailty Index

Longitudinal associations between baseline carbohydrate intake and FI trajectories were assessed in 816 participants with FI data from at least one follow-up visit. Compared to the 208 participants with no follow up data, those participants in the longitudinal analyses had lower baseline FI (*p* < 0.001) and reported lower percentage of energy intake derived from carbohydrates (*p* = 0.004), with higher intakes from MUFA (*p* = 0.01) and SFA (*p* = 0.014) (Appendix A). 

Over an average of 6.5 years of follow up (range 1–13.8 years), FI index increased by 0.009 units per year. To evaluate the association of carbohydate intake on trajectories of FI, we examined the significance of the interaction terms between consumption at baseline and follow-up time in the linear mixed effects model. There was one signficant interaction among women, those in the highest tertile of fiber: carbohydrate experienced significantly less steep increase in FI compared to those in the lowest tertile (Figure 2; Appendix A). For all other models, the interaction term was not significant (Appendix A). 

The main effect of different carbohydrate consumption indexes was significant in several models, suggesting that there is persistent effect of carbohydrate intake on FI throughout the follow-up period considered (Table 2). Specifically, in the combined analysis, the second tertile of total grains, and high tertile of % carbohydrate, and glycemic load was associated with higher FI over time while being in medium and high tertiles of the fiber-to-carbohydrate ratio was associated with lower FI throughout the follow-up period. In men, being in the high tertile of carbohydrate was associated with higher FI throughout the follow-up period. In women, being the medium tertile of glycemic load, total grains, and non-whole grain and being in the high tertile of carbohydrate, glycemic load was associated with higher FI throughout the follow-up period. 

## 4. Discussion

In this population of older men and women, most of the examined indexes of carbohydrate consumption were associated with worse FI except for the fiber-to-carbohydrate ratio where higher consumption was associated with lower FI. The differences persisted throughout the 13-years follow-up period. In women, a higher ratio of fiber-to-carbohydrate was associated with slower progression of FI over time, suggesting that a diet rich in fiber relative to the proportion of carbohydrate can slow the accumulation of deficits related to aging frailty over time.

There has been a large body of work that has examined the link between diet and frailty, particularly focusing on predefined dietary patterns, such as adherence to the Mediterranean diet and Dietary Quality Index-International (DQI-I) [20,21,22]. In general, it was found that higher adherence to a healthy diet that is rich in fruits, vegetables, and whole grains and low in simple sugars and fats is associated with a reduced risk of frailty. While most studies with dietary patterns do not focus solely on carbohydrate consumption, adherence to different healthy dietary patterns represents diets with higher carbohydrate quality. Further, data-guided clustering (a posteriori) analyses show that dietary patterns that represent lower carbohydrate quality are associated with a higher risk of frailty. For example, in a study of 666 participants 60 years and older, the “sugar and fat” dietary pattern was associated with greater progression in frailty over a 3-year follow-up period using a 54-item FI [23]. Taken together, studies on dietary patterns support the premise that a diet rich in high quality carbohydrates may contribute to the prevention of frailty development and progression.

Not all observational studies support the link between carbohydrate consumption and frailty. Two studies that examined macronutrient composition have association between frailty and protein but not carbohydrate consumption. In the first, 1822 participants in the Seniors-ENRICA in Spain demonstrated that overall protein and animal protein intake was significantly associated with incident frailty phenotype over a 3.5-year follow-up period, while no association was observed with simple sugars [23]. Similarly, in a study of 5205 middle-aged participants from the Rotterdam study, protein intake was associated with trajectories of a 38-item FI, while no associations were observed with carbohydrate consumption over a 11-year follow-up period [24]. The lack of association between carbohydrate intake and frailty may be because the study did not fully explore carbohydrate quality and only examined total carbohydrate intake. In this respect, a recent study explored the association between different measures of carbohydrate quality and risk for physical frailty over a 15-year period in 1210 participants of the three-city Bordeaux cohort [12]. In this study, higher intake of simple carbohydrates was associated with a greater risk for developing frailty, particularly in older men. In the same study, however, total carbohydrates and glycemic load was not associated with risk of frailty. These results concur with the present analyses that demonstrated a positive association between higher consumption of carbohydrates and frailty risk, indicating that carbohydrate intake can be detrimental to the aging process. We observed that higher consumption of non-whole grains was positively associated with FI, while whole grains trended towards negative associations with FI, although not statistically significant. The negative association observed between total carbohydrate and FI is likely due to the low quality of carbohydrate consumed based on the greater consumption of non-whole grains relative to whole grains, in the BLSA. 

The mechanisms linking carbohydrate consumption with FI are largely unknown. However, increased carbohydrate intake could influence inflammation or insulin sensitivity which can, in turn, affect skeletal muscle loss or sarcopenia [25,26]. Diets with low carbohydrate quality and high glycemic loads are associated with high post-prandial plasma glucose levels that may contribute chronic inflammation and oxidative stress, and through these mechanisms may increase the risk of developing chronic diseases, including diabetes and cardiovascular disease [27]. In the present analyses, the models were adjusted for fasting glucose and diabetes status, however, it is possible that carbohydrate quality affects the trajectories of glycemic parameters that were not accounted for in these analyses. Future studies are needed to explore metabolic and proteomic changes that are associated with carbohydrate consumption to understand the mechanisms underlying carbohydrate consumption and frailty.

The strength of the present study is the evaluation of different categories of carbohydrate intake in a single aging cohort in relation to trajectories of FI. There are also limitations, namely the fact that BLSA is a cohort of a healthy elderly population with high socioeconomic status and education status which limits the generalizability of these findings. The biases accompanying the use of FFQs in determining the intake of the several carbohydrate categories, although data from the FFQ were validated against multiple 24-h recalls. While the analysis was adjusted for many potential confounders, the possibility of residual confounding cannot be fully excluded.

## 5. Conclusions

The findings of this study suggest that carbohydrate consumption is positively associated with increased frailty risk, while consuming a higher ratio of fiber-to-carbohydrate intake can be beneficial in reducing frailty overtime. Increasing fiber intake can be beneficial for frailty when adhering to dietary patterns with increased intake of healthy food groups, such as fruits and vegetables. In this context, the present findings highlight the potential effect of carbohydrates on aging and aging-related frailty, whilst underlining the need for undertaking holistic approaches in the detailed evaluation of the respective interplay between carbohydrate and fiber intake and overall consumption of various food groups.

## Figures and Tables

**Figure 1 nutrients-14-05072-f001:**
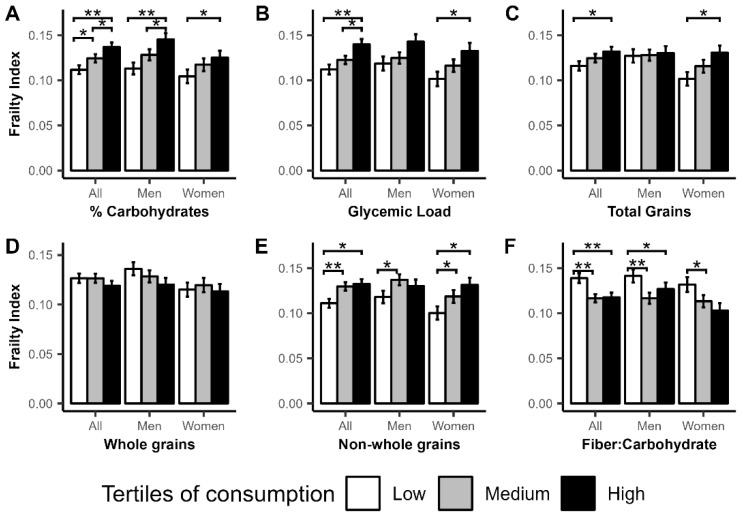
Associations between Frailty Index (FI) and tertiles of consumption of the different carbohydrate categories. Associations between tertiles of carbohydrate consumption and FI were assessed using multiple linear regression adjusted for age, sex, and total energy, BMI, diabetes, percent energy from polyunsaturated fatty acids (PUFA), saturated fatty acid (SFA), monounsaturated fatty acid (MUFA), and total fiber (except for the analysis of fiber-to-carbohydrate ratio). The graph represents mean FI by tertiles of consumption of percent energy from carbohydrate (**A**), glycemic load (**B**), total grains (**C**), whole grains (**D**), non-whole grains (**E**), and ratio of fiber to total carbohydrate (**F**). ** *p* ≤ 0.001, * *p* ≤ 0.05.

**Figure 2 nutrients-14-05072-f002:**
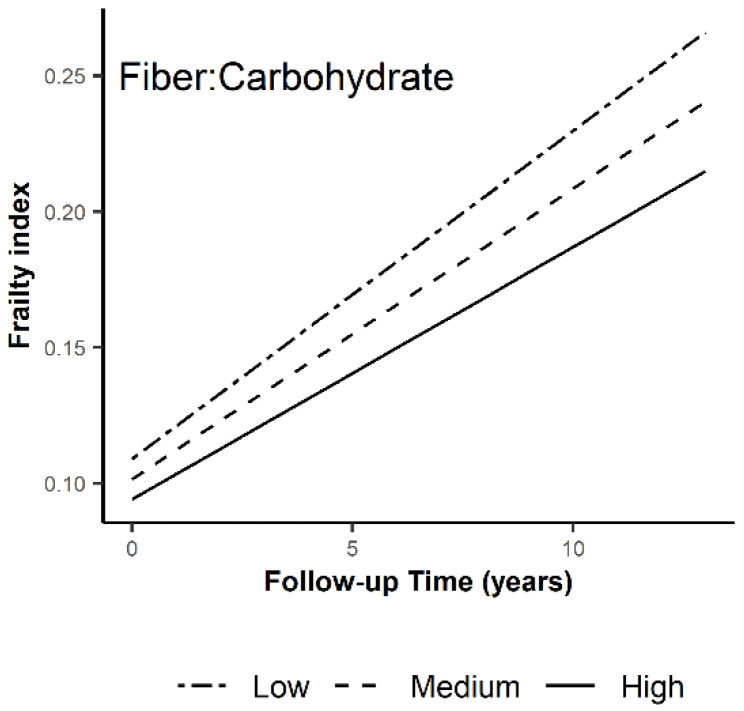
Trajectory of FI in the tertiles of fiber-to-carbohydrate ratio in women. Association between baseline consumption of fiber-to-carbohydrate ratio was evaluated using linear mixed effects model. Among women, those in the highest tertile of fiber: carbohydrate experience significantly less steep increase in FI compared to those in the lowest tertile.

**Table 1 nutrients-14-05072-t001:** Descriptive characteristics of the Baltimore Longitudinal Study of Aging.

	Combined		Women		Men		*p*
*n*	1024		523		501		
Frailty Index	0.11	(0.08)	0.11	(0.08)	0.11	(0.08)	0.653
Age (years)	74	(7.86)	73.3	(8.03)	74.72	(7.62)	0.004
Sex (%men)	501	(48.90)	0	(0.00)	501	(100.00)	
Diabetes	134	(13.10)	48	(9.20)	86	(17.20)	<0.001 *
Fasting glucose (mg/dL)	92.03	(18.30)	89.48	(15.11)	94.69	(20.81)	<0.001 *
HBA1C (%)	5.85	(0.59)	5.87	(0.56)	5.83	(0.63)	0.378 *
BMI (kg/m^2^)	26.93	(4.59)	26.63	(4.94)	27.23	(4.18)	0.005 *
Total energy (kcal/day)	1953.6	(672.69)	1805.94	(606.92)	2107.76	(703.33)	<0.001 *
Glycemic load (g/day) **	107.54	(42.78)	99.32	(39.29)	116.12	(44.59)	<0.001 *
Total grain (serv/day)	6.01	(2.65)	5.41	(2.28)	6.64	(2.86)	<0.001 *
Whole grains (serv/day)	1.81	(1.14)	1.73	(1.05)	1.9	(1.22)	0.009 *
Non-whole grains (serv/day)	4.2	(2.18)	3.69	(1.74)	4.73	(2.44)	<0.001 *
Fiber: Carbohydrate	0.1	(0.02)	0.1	(0.03)	0.09	(0.02)	<0.001 *
Fiber (g/day)	20.92	(7.97)	20.57	(8.10)	21.27	(7.83)	0.161 *
% energy Carbohydrate	46.28	(7.32)	46.78	(7.16)	45.75	(7.45)	0.003 *
% energy PUFA	7.91	(1.73)	8.12	(1.67)	7.7	(1.77)	<0.001 *
% energy MUFA	12.72	(2.19)	12.84	(2.07)	12.6	(2.31)	0.175 *
% energy SFA	10.99	(2.36)	11	(2.25)	10.97	(2.47)	0.990 *

Data represent mean (SD) for continuous variables or *n* (%) for categorical variables. * Age-adjusted differences by sex. ** The glycemic load: mean net carbohydrate intake (g/day) multiplied by the glycemic index divided by 100.

**Table 2 nutrients-14-05072-t002:** Main effects of carbohydrate consumption on trajectories of frailty index.

		All			Men			Women	
	β *	(SE)	*p*	β *	(SE)	*p*	β *	(SE)	*p*
% Carbohydrate									
Med	0.005	(0.005)	0.291	0.008	(0.007)	0.256	0.003	(0.007)	0.692
High	0.020	(0.006)	0.001	0.025	(0.008)	0.003	0.015	(0.009)	0.091
Glycemic Load									
Med	0.008	(0.006)	0.147	−0.001	(0.008)	0.945	0.016	(0.007)	0.039
High	0.017	(0.008)	0.041	0.018	(0.012)	0.132	0.014	(0.011)	0.221
Total Grains									
Med	0.006	(0.005)	0.279	−0.006	(0.008)	0.471	0.013	(0.007)	0.067
High	0.004	(0.007)	0.523	−0.007	(0.010)	0.487	0.016	(0.009)	0.088
Whole Grains									
Med	0.006	(0.005)	0.261	−0.001	(0.007)	0.923	0.010	(0.007)	0.168
High	−0.007	(0.006)	0.191	−0.017	(0.008)	0.037	0.003	(0.008)	0.679
Non-whole grains									
Med	0.011	(0.005)	0.031	0.004	(0.007)	0.554	0.017	(0.007)	0.015
High	0.012	(0.007)	0.059	0.001	(0.009)	0.947	0.025	(0.009)	0.007
Fiber: Carbohydrate									
Med	−0.013	(0.005)	0.008	−0.013	(0.007)	0.065	−0.014	(0.007)	0.044
High	−0.010	(0.005)	0.050	−0.007	(0.007)	0.365	−0.015	(0.007)	0.035

* data represent beta estimates for the main effects of tertiles of carbohydrate consumption in a mixed effects model.

## Data Availability

Data from Baltimore Longitudinal Study of Aging are available through submission of research proposal through https://www.blsa.nih.gov/.

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
