# Peer review of "Quality Specific Associations of Carbohydrate Consumption and Frailty Index"

_nutrients, 2022, doi:10.3390/nu14235072_

Round 1

Reviewer 1 Report

the authors aimed to study the association between the quality of carbohydrate consumation and the risk of frailty. they created a 43-item Frailty Index (FI) for evaluating risk of frailty.

I want to know is:

is the FI item being universally accepted?

is the FI item get consistency check?

did the author observe and compare clinical adverse event ?

Author Response

We thank the reviewer for the comments. We have replied to the inquiries below in bolded text. Regarding corrections for English language and style, I assure the reviewers and editor that the majority of the authors are native English speakers with extensive experience in writing manuscripts. We will nevertheless revise the manuscript for any grammatical errors and legibility.

is the FI item being universally accepted?

>> Yes, FI items can vary, however, many frailty index include similar clinical variables that we have included in our construct. We follow the instructions by Searle et al as described in the introduction (line 38).

is the FI item get consistency check?

>>>In terms of comparing the FI in the BLSA to other studies, we have selected variables that were selected items that are consistent with the InCHIANTI study (PMID 32006385). In addition, many of the frailty index developed by other cohorts have similar clinical and behavioral traits that were included in the BLSA FI.

did the author observe and compare clinical adverse event ?

>>We thank the reviewers for this suggestion. The aim of the project was to examine the association between carbohydrate consumption and frailty index. The beauty of frailty index is that the index includes many of the important clinical aging outcomes including chronic diseases, mobility disability, and cognitive function. The data suggest that carbohydrate consumption has a global impact on the accumulation of these important clinical events. However, in the future we could explore individual clinical outcome in addition to others such as mortality.

Reviewer 2 Report

In my opinion, the manuscript was well prepared

  • 1) The Introduction Section explains the design of the study. The Authors well justify the research topic. 

  • 2) The study was carried out without methodological errors. 

  • 3) The Descriptions of the results were correct. 

  • 4) The presented figures and table were prepared precisely and also legible. 

  • 5) The Discussion Section included the accurate reference of the results obtained to the studies of other authors.  

  • 6) The Conclusions were well formulated.

Author Response

We thank the reviewer for the kind and generous feedback.